# On the Relationship between Feature Selection Metrics and Accuracy

**DOI:** 10.3390/e25121646

**Published:** 2023-12-11

**Authors:** Elise Epstein, Naren Nallapareddy, Soumya Ray

**Affiliations:** Department of Computer and Data Sciences, Case Western Reserve University, Cleveland, OH 44106, USA; nxn151@case.edu

**Keywords:** feature selection, model selection, decision trees

## Abstract

Feature selection metrics are commonly used in the machine learning pipeline to rank and select features before creating a predictive model. While many different metrics have been proposed for feature selection, final models are often evaluated by accuracy. In this paper, we consider the relationship between common feature selection metrics and accuracy. In particular, we focus on *misorderings*: cases where a feature selection metric may rank features differently than accuracy would. We analytically investigate the frequency of misordering for a variety of feature selection metrics as a function of parameters that represent how a feature partitions the data. Our analysis reveals that different metrics have systematic differences in how likely they are to misorder features which can happen over a wide range of partition parameters. We then perform an empirical evaluation with different feature selection metrics on several real-world datasets to measure misordering. Our empirical results generally match our analytical results, illustrating that misordering features happens in practice and can provide some insight into the performance of feature selection metrics.

## 1. Introduction

In supervised machine learning, many algorithms assume a feature-set representation of the data in which each feature represents a characteristic of the underlying subject being analyzed. However, not all features have the same effect on prediction. When predicting if an animal is a lion, whether the animal has sharp teeth is likely a better indicator than whether the animal has eyes. As the number of features increases, the difficulty of learning a concept increases significantly [1]. This phenomenon is often referred to as the “curse of dimensionality” [1]. Together with this, the chance of overfitting, that is, producing a model that fits the training data very well but does poorly overall, increases as well. In order to mitigate these issues, a common strategy is to filter the feature set that is to try to identify “relevant” features. Relevant features are those that are hypothesized to be used by the underlying target concept we seek. Once a set of candidate-relevant features is identified, concepts may be learned with these features alone. This speeds up the search process and results in the learned concepts being less likely to overfit since they contain fewer (or no) irrelevant features.

After selecting a relevant feature subset, predictive models are built. Such models often optimize the 0–1 loss and are evaluated according to accuracy. While many alternative losses and evaluation metrics exist, the popularity of the 0–1 loss for optimization and accuracy as a metric is due to the simple and intuitive nature of these functions [2]. However, the 0–1 loss is itself rarely used for feature selection. Possible reasons are that finding a set of features that optimizes the 0–1 loss is not tractable in the worst case [3,4,5,6], and optimizing this loss alone may not necessarily yield a small set of features. Instead, feature selection filters use a variety of metrics such as information gain [7,8,9,10,11] and Gini index [12]. Metrics such as these are typically well-correlated with accuracy but do not always produce the same result. Inthis paper, our interest is to specifically analyze and gain insight into cases when feature selection metrics rank features differently than accuracy would. This is of interest because it may provide insight into the downstream performance of a predictive model optimizing the 0–1 loss or evaluated using accuracy as a metric.

As a simple example, consider a toy dataset in Table 1. The task is to learn whether an animal is a lion. If we were to apply information gain as a filter to select the top feature, *Loud?* will be selected with a gain of 0.1. However, a decision stump classifier using either *Has Fur?* or *Lazy?* would have a higher accuracy (0.67 vs. 0.56). These features have a lower information gain than *Loud?* (0.07 and 0.09 respectively). In this case, we would say that the two pairs (*Has Fur?*, *Loud?*) and (*Lazy?*, *Loud?*) are misordered by information gain relative to accuracy. A misordering occurs when, without loss of generality, there are two features where feature A has a higher feature selection metric value than feature B, but feature B has a higher accuracy when used as a decision stump classifier. Of course, in this toy example, the impact of misordering may not be very significant. But in real-world tasks, with many features, it is possible that feature selection metrics that are more prone to misordering may lead to poorer performance in downstream models.

The fundamental reason why misorderings may occur is because there is not a one-to-one correspondence between feature selection metrics and accuracy. This was established in prior work [13] which derived bounds on the classification accuracy given information gain and vice versa for the binary classification case. However, the implications of those results in terms of feature selection are not well understood, to our knowledge. Our work is a step to fill this gap.

We make the following assumptions for this work:For each feature selection metric, a higher value corresponds to a more relevant feature for the prediction task.Feature selection metrics are evaluated in the context of binary classification, where one of two class labels are associated with each example [14]. While our analysis can be generalized to multiclass tasks, our results in this work use data from binary classification.We consider each feature to make a binary partition of the data in order to simplify the analysis. This does not impose a loss of generality because continuous or nominal features with many values can still be used to perform binary partitions [12].We assume no feature values are missing for the purpose of analytical comparisons.

In the following sections, we first introduce some definitions and notations and then define the conditions under which misordering can happen for for feature selection metrics, in a space parameterized by the characteristics of a partition induced by a feature. Next, we visualize the probability of misordering both locally and globally in this space. Finally, we evaluate a number of feature selection metrics on real-world data to understand the prevalence of misordering in reality.

## 2. Conditions for Misordering

We define each input example as $x_{i,j}$ drawn from an input space X. Here, *i* represents the index of the example in dataset D, and *j* denotes the feature index of the example. Each example has an associated class $y_{k}$ in the output space Y, where *k* represents the class index.
(1)$$D\equiv \{\left(x_{i,j},y_{k}\right)|i\in \left(1,\ldots ,N\right),j\in \left(1,\ldots ,F\right),k\in \left(1,\ldots ,C\right)\}$$

In this representation, *N* denotes the number of samples, *F* represents the number of features each sample has and *C* is the total number of classes in the dataset D. In the rest of the paper we focus on binary classification, which can be characterized using two classes.

We further define a partition as a binary split in the dataset, based on a feature *j*. We identify two feature values corresponding to the canonical feature values 0 and 1, each containing a mixture of positive and negative examples. Feature value 0 has *a* negative and *b* positive examples, while feature value 1 has *c* negative and *d* positive examples. Mathematically, this can be expressed as partition(j)={0:[a,b],1:[c,d]} where partition(j) represents the partition on feature *j*.

We observe that the variables a,b,c, and *d* can completely represent both the partitions as well as the original dataset D. However, they are not independent. To address this, we introduce two alternate variables *p* and *q*. Here, *p* represents the fraction of examples in the first partition, and *q* represents the fraction of examples that have positive labels out of examples in the first partition. Additionally, *m* represents the fraction of negative examples in the dataset D. Using *p*, *q*, and *m*, we can completely represent any partition as below:(2)$$partition\left(j\right)=\{0:\left[Np_{j}\left(1-q_{j}\right),Np_{j}q_{j}\right],1:\left[Nm-Np_{j}\left(1-q_{j}\right),N\left(1-m\right)-Np_{j}q_{j}\right]\}$$

Using this change of variables (Equation (Equation 2)), we can express the accuracy of a partition as:(3)$$\begin{array}{c} Acc\left(p,q,m\right)=\left\{\begin{array}{cc} p(1-q)+\max(m-p(1-q),(1-m)-pq) & \mathrm{if}\;q<0.5\\ pq+\max(m-p(1-q),(1-m)-pq) & \mathrm{if}\;q\geq 0.5 \end{array}\right. \end{array}$$
In these equations max(.) represents the max operator. We only provide the final expression for sake of brevity.

### 2.1. Feature Selection Metrics Used

In this paper, we focus primarily on information gain, Gini index, Hellinger distance, and Bhattacharyya distance to perform an analytical comparison. An empirical comparison of various real datasets with additional feature selection methods is presented in Section 4.

#### 2.1.1. Information Gain

Information gain is a widely used feature selection method [15]. It measures the reduction in uncertainty for predicting the target class when considering feature *j*. Mathematically, it can be expressed as IG(Y,j)=H(Y)−H(Y|j), where H(.) denotes the entropy of a random variable, *Y* represents the output random variables and *j* represents the feature.

Information gain can be written in terms of the variables *p*, *q* and *m* as
(4)$$\begin{array}{cc} IG(p,q,m) & =\left(1-m\right)\log_{2}\left(\frac{1}{1-m}\right)+m\log_{2}\left(\frac{1}{m}\right)\\ \; & -[p\left(1-q\right)\log_{2}\left(\frac{1}{1-q}\right)+pq\log_{2}\left(\frac{1}{q}\right)\\ \; & +\left(m-p\left(1-q\right)\right)\log_{2}\left(\frac{1-p}{m-p(1-q)}\right)\\ \; & +\left(\left(1-m\right)-pq\right)\log_{2}\left(\frac{1-p}{(1-m)-pq}\right)] \end{array}$$

#### 2.1.2. Gini Index

The Gini index is another important filter method, which can be expressed as $Gini\left(Y\right)=1-\sum_{k=0}^{1}P_{k}^{2}$ [16]. Here, *Y* is a discrete output random variable, and $P_{k}$ is the probability of a sample $x_{i}$ being classified into class *k*. Essentially, the Gini index measures the impurity of a given dataset. Analogous to information gain, the difference between the Gini index of the dataset D and the subsets resulting from splitting using different features 1,…,F are used to rank features. The Gini index can be expressed in terms of the variables *p*, *q*, and *m* as:(5)$$Gini\left(p,q,m\right)=\frac{2p(1-m-q^{2})}{1-p}$$

#### 2.1.3. Hellinger Distance

Hellinger distance is a statistical measure that quantifies the similarity between two probability distributions. It can be defined in terms of two discrete random variables *X* and *Y* as $HD\left(X,Y\right)=\frac{1}{\sqrt{2}}\sqrt{\sum_{k=1}^{C}{\left(\sqrt{x_{k}}-\sqrt{y_{k}}\right)}^{2}}$ [17]. To utilize the Hellinger distance as a feature selection metric, we calculate the partition weighted average between the Hellinger distance of the original dataset and the partitions. Our goal in feature selection is to choose the partition with maximal distributional distance from the original dataset. Hellinger distance can be expressed in terms of the variables *p*, *q*, and *m* as:(6)$$\begin{array}{cc} HD(p,q,m) & =\frac{p(1-m-q)}{\left(1-p\right)}\\ \; & -p\left(\left(1-m-q\right)+\frac{p(1-m-q)}{\left(1-p\right)}\right) \end{array}$$

#### 2.1.4. Bhattacharyya Distance

Bhattacharyya distance is another measure of similarity between two probability distributions. Unlike Hellinger distance, Bhattacharyya distance does not obey the triangle inequality, therefore, it is not considered a true distance. It is defined as $BD\left(X,Y\right)=-\ln\sum_{k=1}^{C}x_{k}y_{k}$ where *X* and *Y* are two discrete random variables defined on the same space, ln denotes the natural logarithm and BD(.) is the Bhattacharyya distance function [18]. Analogous to Hellinger distance, for its use in feature selection, we compute the partition weighted average between the Bhattacharyya distance of the original dataset with the partitions [19]. Bhattacharyya distance can also be expressed in terms of variables *p*, *q*, and *m* as:(7)$$\begin{array}{cc} BD(p,q,m) & =p\ln(m^{2}-m+q^{2}-q)\\ \; & -\left(1-p\right)\ln(\frac{\left(m-1\right)(m\left(p^{2}-2p+2\right)}{{\left(1-p\right)}^{2}}\\ \; & +\frac{pq\left(p-2m+1\right)-p^{2}q^{2}}{{\left(1-p\right)}^{2}}) \end{array}$$

In order to compute the misorderings for a given set of partitions with variables ($p_{0}$, $q_{0}$) belonging to partition 0, and ($p_{1}$, $q_{1}$) belonging to partition 1 and the fraction of samples in the dataset D given by *m*. Using these variables we can calculate the misorderings as
(8)$$\begin{array}{cc} Misorder_{FS}\left(p_{0},q_{0},p_{1},q_{1},m\right)= & FS\left(p_{0},q_{0},m\right)<FS\left(p_{1},q_{1},m\right)\;\&\&\\ & Acc\left(p_{0},q_{0},m\right)>Acc\left(p_{1},q_{1},m\right)\;∥\\ \; & FS\left(p_{0},q_{0},m\right)>FS\left(p_{1},q_{1},m\right)\;\&\&\\ & Acc\left(p_{0},q_{0},m\right)<Acc\left(p_{1},q_{1},m\right) \end{array}$$
Here, FS(.) is a feature selection function that is one of IG(.), Gini(.), HD(.) or BD(.). In the expression, && represents the logical *and* operator, similarly ∥ represents the logical *or* operator.

## 3. Analytical Comparison and Discussion

In this section, we analyze Equation (Equation 8) derived in the previous section to understand to what extent each approach is prone to misordering features. To conduct this we study the behavior of the inequality as a function of its inputs graphically for each feature selection metric.

### 3.1. A Local View of Misordering

Consider some partition (p,q) for a given *m*. How prone is the point (p,q) to a misordering? For a given feature selector, consider the unit vector along the derivative of the selector with respect to (p,q) and the unit vector along the derivative of accuracy at (p,q). Our hypothesis is that the more aligned these vectors are, the less prone (p,q) is to misordering. The intuition is that if these vectors are aligned in the local environment around (p,q), then both accuracy and the metric are changing in the same way. As they become misaligned, it becomes increasingly possible for a misordering to happen. To quantify this effect, the alignment between the direction of the two vectors is calculated using the cosine of the angle θ between them. We compute 1−cos(θ) to quantify misalignment between the two vectors. We then plot heat maps of the misalignment in (p,q) space using small steps in *p* and *q* parameters spanning all possible values of *p*, and *q*. In our computation, gradient vectors are generated using automatic differentiation to overcome any finite step effects. The results are shown in Figure 1.

In the heat maps shown in Figure 1, bright yellow regions represent significant misalignment between gradient vectors of the corresponding feature selection metric and that of accuracy. The darkest blue region denotes a configuration of (p,q) that is unachievable for the given *m*). For m=0.5 (Figure 1a–d), we observe that Bhattacharyya distance (Figure 1d) qualitatively has the highest misalignment with accuracy, while Hellinger distance (Figure 1c) has qualitatively the lowest misalignment with accuracy. Conversely, for skewed datasets (Figure 1e–h) qualitatively Hellinger distance (Figure 1g) has the highest misalignment with accuracy.

Considering the graphs for different values of *m*, we observe that regions of significant misalignment grow as *m* increases (the class distribution becomes more skewed) for all metrics. This indicates that the problem of misordering should become more extensive as *m* grows.

We identify distinct and consistent regions on the heatmaps across all feature selection metrics that have significant dissimilarity with accuracy as a function of *m*. Such observations indicate it may be possible to characterize these regions and predict when misordering is likely to happen.

### 3.2. A Global View of Misordering

The previous analysis describes the behavior of feature selection metrics around a specific (p,q) point. In this subsection, we examine the distribution of misordering probability as a function of the individual parameters *p* and *q* for different values of *m*. To do this, we will average over all other parameters. That is, for a fixed *m*, we will consider all pairs of partitions $\left(p_{0},q\right)$ and $\left(p_{1},q_{1}\right)$ for a given value of *q*. The space of all such partitions will be finely sampled. This approach enables us to calculate the misordering probability as fraction of misorderings over the space of all possible combinations of possible partitions. Note that since the order of the partition pair does not matter, we also include all $\left(p_{0},q_{0}\right)$ and $\left(p_{1},q\right)$ pairs in the analysis for a given *q*. The results are shown in Figure 2 as a function of *p* and Figure 3 as a function of *q*.

From Figure 3, we make some key observations. Among the feature selection metrics chosen, Bhattacharyya distance, Information gain and Gini have consistent ordering for both balanced (Figure 3a) as well as skewed datasets (Figure 3b). Among these three metrics, Bhattacharyya distance has the highest misordering probability followed by Information gain and Gini, respectively. Particularly, Gini maintains a relatively lower misordering probability across both balanced as well as skewed datasets suggesting resilience to variation in *m*.

In contrast, Hellinger distance displays interesting behavior. It has the lowest misordering probability for m=0.5 at the same time has the highest misordering probability for a higher value of *m*. This suggests that Hellinger distance is particularly sensitive to skewness of dataset. This analysis is consistent with our observations in the previous section about the local behavior of this metric.

Examining the misordering probability plotted against the parameter *p* in Figure 2, we can make analogous statements to misordering against the parameter *q*. Information gain, Gini and Bhattacharyya distance have consistent ordering in misordering probability across both balanced as well as skewed datasets. Hellinger distance maintains similar divergent behavior between balanced and skewed datasets.

However, in contrast to Figure 3, the plots against the parameter *p* are flatter, which suggests there is less sensitivity to a fraction of samples in the first partition.

To further highlight the dependence of Hellinger distance on *m* we conduct an additional experiment. We vary *m*, using m=0.5, m=0.55, m=0.6, and m=0.8. This allows us to illustrate the crossover point for Hellinger distance, where the misordering probability of Hellinger distance reverts from having the lowest misordering probability to the highest misordering probability.

From Figure 4 we observe that there is a crossover between Figure 4b and Figure 4c. At this transition point, Hellinger distance has a misordering probability comparable to other feature selection metrics. This suggests that by knowing the skewness of the data distribution, we can make an informed choice to suppress misordering due to feature selection.

In order to verify that the theoretical results shown in Figure 3 are consistent with generated partitions, we created synthetic datasets with the same class skews. We plot the misordering rate of information gain against *q* as shown in Figure 5. We randomly generated 5000 partitions. The partitions are grouped into groups of *q* values such that there is a minimum of 30 partitions for each group. For Figure 5a, this results in bins of *q* values of 0.02, while for Figure 5b this results in bins of 0.025. From Figure 3 and Figure 5 we see similar shapes for information gain, which suggests that the theoretical results in this section will apply to real data.

### 3.3. Is Misordering Confined to Narrow Combinations of Error Rates and Feature Selection Metrics?

An important question arises when discussing misorderings—what is the extent of feature selection metrics and accuracy values that can result in misordering? One might naturally assume that misorderings would occur within a confined region of the space of the feature selection metric and accuracy. If true, this would suggest that the issue of misordering is not a widespread phenomenon.

In this experiment, we aim to determine the range of values of the feature selection metric and accuracy that can lead to misordering. To determine this, we sample across the complete space of *p* and *q* values. This enables us to examine a range of partitions that can arise for a binary classification task.

From Figure 6, we observe that the values of information gain, Gini index, Hellinger distance and Bhattacharyya distance and rate span a significant region. This may be due to the fact that misordered pairs arise from the relationship between two partitions, not just a single partition, meaning the misordering region is unlikely to be constrained to a small area in the space of any given feature selection metric and error rate.

We note that as the distribution of negative and positive samples becomes more skewed, the region representing misordering shrinks. This occurs because fewer positive examples in a dataset result in a smaller range of possible error values as well as information gain values.

We also notice that the extent of error values for skewed distribution (as shown in Figure 6e–h) is smaller. This can be attributed to the fact that the maximum possible error in a skewed dataset is determined by the percent of the data in the minority class. In the case of m=0.8, the skewed distribution of negative to positive samples is 0.2.

Misordering of partitions can occur in a wide region of space spanned by the feature selection metric and the error rate. This means that misordering is not localized to specific feature selection metrics or accuracy values.

### 3.4. Is Misordering Primarily Due to Numerical Issues?

Another possible explanation for the observed misordering could be attributed to rounding errors while ranking features. This is relevant when a pair of partitions exhibits very close values in terms of either information gain or error rate. To verify this hypothesis, we examine the relationship by plotting the difference in various feature selection metrics against the difference in error rate between two partitions. Similar to the previous experiment, we sample across the complete space of *p* and *q* values. This enables us to consider a range of partitions that can result in misordering. Note, that unlike Section 3.1, the pairs of partitions we consider here can have very different (p,q) values.

In Figure 7, we illustrate the relationships between the difference in feature selection metrics and the difference in error. It is important to note that there is a symmetric region of misordered partitions that spans the negative x-y axis in the fourth quadrant, this region is omitted in the illustration for the sake of simplicity.

We note that the extent of values for the difference in the feature selection metric and the difference in error is significant and finite. This observation indicates that misordering *cannot* be attributed exclusively to rounding errors when a pair of partitions have closely aligned values in either information gain or error.

We further observe a notable difference between skewed and balanced datasets. Specifically, we observe that for skewed datasets (as depicted in Figure 7e–h), the extent of difference in the feature selection metric and error is larger than for the balanced dataset (shown in Figure 7a–d). This suggests a greater probability of misordering in skewed datasets compared to balanced datasets. This is a likely supporting cause for the greater probability of misordering we observed in our prior analysis.

## 4. Empirical Comparison and Discussion

In the previous section, we considered the probability of misordering analytically. However, this does not establish that misordering can happen in real data, or if it does, what is its prevalence. In this section, we study this question by comparing 13 real-world datasets of varying sizes. Three of these datasets (Arcene, Gisette, and Madelon) are used in the Neural Information Processing Systems 2003 Feature Selection Challenge [20]. We also use biological/text datasets which include ALLAML, Colon, GLI_85, Leukemia, Prostate_GE, SMK_CAN_187, and BASEHOCK which can be retrieved at [21]. The last category of datasets we use are SONAR, Ionosphere, and Congressional Voting from [22]. The characteristics of all of these datasets can be found in Table 2. We chose datasets that span with varying numbers and types of features, class ratios and numbers of examples to get a holistic view of the misordering problem.

### 4.1. Misordering on Real-World Datasets

We evaluated misordering frequency on the selected datasets using three feature selection metrics information gain, the Gini index and Gain Ratio [15,23]. The gain Ratio is a modification of information gain that attempts to calibrate features that have a large variance in the number of values. Although in our analytical comparisons (Section 3) we have considered features to have two values, real data can have many. To rank such features we must consider all their values. The Gini index generalizes well to such cases. On the other hand, information gain prefers features with a large number of values. To counterbalance this, the Gain Ratio normalizes information gain with the entropy of the feature, GR(X)=IG(X)/H(X). The core idea is that a feature with a large number of values will also have high entropy, so dividing by the entropy will make such a feature less likely to be ranked highly.

The misordering rates for all datasets and these metrics are provided in Table 3. Based on this table, we first observe that misordering does indeed happen in real datasets for all metrics we considered. It ranges from 1.8% (Voting/Gain Ratio) to 14.3% (Madelon/Gain Ratio). In the majority of cases, about 6% of the feature pairs are misordered by most of these metrics. Second, when comparing information gain with the Gini index, across all datasets, the Gini index has a lower or equal misordering rate to information gain. This is consistent with our analysis above, where we saw the same behavior. The gain Ratio does not show a clear pattern. In some datasets, its misordering rates are lower than others, and in some, they are the highest.

One interesting observation from this data is that the misordering rates for information gain and the Gini index seem to be relatively stable as the class proportion changes. From the analytical results in Figure 3 and Figure 4, we observe that the misordering rates tend to increase on average as the class proportions become more unbalanced. However, this trend is not reflected in our real-world datasets. Of course, our analytical curves are averages, so this behavior is not ruled out by those results. Second, the most skewed datasets in our comparison are also among the smallest. Such small samples may bias estimates of information gain, Gini, or accuracy which could affect the misordering rate differently.

Another important observation from the empirical results in Section 3 is the effect of class imbalance on the misordering rate of Hellinger distance. The results in Figure 4 show that Hellinger distance has the lowest misordering rate for class-balanced datasets, and the rate increases with class imbalance. Using datasets from Figure Table 2, we plot Hellinger distance vs. class skew in Figure 8a. These results verify the analytical results on real data. Using a linear fit to the data gives $R^{2}=0.71$ which suggests our empirical observations are consistent with our observations from real datasets. For comparison, we examined the percent misordering due to information gain against the class skew in Figure 8b. Here we find no significant correlation between the linear model and data points, which corresponds to our observations regarding information gain.

A third relevant hypothesis is that the number of unique partitions in a dataset affects the misordering rate. This is because the more partitions there are, the more pairs there are to compare, resulting in a higher chance that two partitions will have a similar enough value of information gain and error to cause misordering. Note that the number of *features* in a dataset is not informative since misordering rates are computed by comparing unique partitions, not features. The number of unique partitions for a dataset is computed by creating all possible partitions through splitting features and then removing all duplicate partitions. This constitutes a set of unique partitions that we use to compute misordering rates. We plot the number of features and misordering rates for information gain for the datasets in Table 2 and Figure 9. The trend line has a positive slope ($R^{2}=0.48$), which is consistent with our hypothesis. However, the slope is inherently small due to the orders of magnitude difference between the values of the axes.

To summarize, our evaluation of real data is broadly consistent with our analytical observations. Different methods do have different misordering rates on different datasets. The ordering of misordering rates on average is broadly consistent across analytical and real-data results, so that, for example, the misordering rate for the Gini index is generally lower than that of information gain. We also observe that Hellinger distance does have an increasing misordering rate as class skew increases, as predicted by our analytical experiments, and the number of features generally increases the misordering rate.

### 4.2. Potential Misordering in Other Feature Selection Metrics

The literature on feature selection metrics is vast. One could ask if other metrics exhibit the same characteristics as those we have evaluated. While it is infeasible for us to implement and test every such method, we attempt to indirectly answer this question. We collect a number of alternative feature selection metrics and evaluate the correlation between their performance as reported in the literature with information gain. Our hypothesis is that methods that produce downstream classifiers that are highly correlated with those produced by information gain will exhibit the same pattern of misordering as information gain.

The methods we evaluate are as follows:**Fisher score** [24] is a filter method that determines the ratio of the separation between classes and the variance within classes for each feature. The idea is to find a subset of features such that the distances between data points of different classes are as large as possible and the distances between data points of the same class are as small as possible.**CFS (Correlation Based Feature Selection)** [25] is a multivariate filter algorithm that ranks subsets of features based on a correlation-based heuristic evaluation function that tends toward subsets that contain features that are correlated with the class but not correlated with each other [7].**Cons (Consistency Based Filter)** [26] is a filter method that measures how much consistency there is in the training data when considering a subset of features [27]. A consistency measure is defined by how often feature values for examples match and the class labels match. A set of features is inconsistent if all of the feature values match, but the class labels differ [28].**Chi-squared statistic** [29] measures the divergence between the observed and expected distribution of a feature and tests whether the distribution of a feature varies between groups. A higher statistic value indicates a feature better for prediction [29].**ReliefF** [30] randomly samples an example from the data and then determines the two nearest neighbors, where one has the same class label and the other has the opposite class label. These nearest neighbors are determined by using the metric of the *p*-dimensional Euclidean distance where *p* is the number of features in the data. The feature values of these neighbors are compared to the example that was sampled and used to update the relevance of each feature to the label. The idea is that a feature that predicts the label well should be able to separate examples with different class labels and predict the same label for examples from the same class [7].**mRMR** [31] chooses features based on the relevance with the target class and also picks features that are minimally redundant with each other. The optimization criteria of minimum redundancy and maximum relevance are based on mutual information [7]. The mutual information between two features represents the dependence between the two variables and thus is used as a measure of redundancy, while the average mutual information over the set of features is used as a measure of relevance. In order to find the minimum redundancy-maximum relevance set of features, these two criteria are optimized simultaneously, which can be done through a variety of heuristics [31].**FCBF (fast correlation-based filter)** [32] first selects a set of features highly correlated with the class label using a metric called Symmetric Uncertainty. Symmetric Uncertainty is the ratio between the information gain and the entropy for two features. Then, three heuristics that keep more relevant features and remove redundant features are applied [7].**INTERACT** [33] also uses Symmetric Uncertainty, the metric used within the FCBF algorithm, but also has a component that focuses on consistency as first mentioned in the Cons metric [7].**LASSO** [34] is an embedded feature selection method for linear models that penalizes the size of the model coefficients and pushes them to be zero [34,35]. Features with zero (or near-zero) coefficients are not selected.**SVM-RFE** [36] is a wrapper method for Support Vector Machine (SVM) classifiers that iteratively trains with the current features and then removes the least important feature based on the trained SVM [7].

To compare the performance of these methods with information gain, we compile results from a large number of papers in the literature (see Appendix A, Table A1 for a list of papers for each method). Each of these papers reports accuracy values for using information gain with a specific model (ex. decision trees) on multiple datasets and then reports the accuracy values for using another feature selection method with the same model and same dataset. We compile all of the information gain accuracy values into one vector and all of the accuracy values for the other feature selection method (ex. chi-squared) in another. For instance, in [37], one experiment was done where the following values were reported: (1) the accuracy using information gained with their Cornell sentiment polarity dataset and training it with a Support Vector Machine and (2) the accuracy using the chi-squared statistic with the same data and training it with a Support Vector Machine. These two entries are paired in the respective performance vectors so that they will be compared when we carry out a correlation.

To ensure that correlation values are valid, the elements within each vector need to be independent. Thus, we avoided using similar classifiers for the same datasets (More details about the specific values used are in the Appendix A and [38]). In order to measure correlation, we use Pearson’s correlation coefficient (*r*) [39]. *X*, *Y* are random variables, if $x_{j},y_{j}$, j=1,2,…,k are sampled, and $\bar{x}$, $\bar{y}$ are the sample means, the correlation coefficient is computed as:(9)$$r=\frac{\sum_{j=1}^{k}\left(x_{j}-\bar{x}\right)\left(y_{j}-\bar{y}\right)}{\sqrt{\sum_{j=1}^{k}{\left(x_{j}-\bar{x}\right)}^{2}\sum_{j=1}^{k}{\left(y_{j}-\bar{y}\right)}^{2}}}$$
The results of this analysis are shown in Table 4.

From the table, we see that nearly every approach we evaluate has performance that is significantly correlated with information gain. The only exceptions (to some extent) are the LASSO and SVM-RFE methods. This is not surprising because these methods are embedded and wrapper methods that perform feature selection in the context of a classifier. All other methods are filters that are very highly correlated with information gain. While this is indirect evidence, it suggests that the results of this paper with information gain likely apply to other feature selection metrics to a similar extent.

### 4.3. Impact of Misordering on Adaboost

In this section and the next, we consider the impact that misordering may have on feature selection and boosting. Adaboost (Adaptive Boosting) [40] is an ensemble algorithm that combines an ensemble of weak learners, into a strong learner. It adaptively weighs data points to generate successive weak learners. Decision stumps using information gain are a common choice as a weak learner for this approach. The theory of boosting suggests that [41], as long as a weak learner that minimizes error can be selected in each iteration, an upper bound of the error of the ensemble will decrease exponentially on the training set. Since we know that information gain can misorder features, thereby not necessarily selecting the lowest error stump, this motivates us to investigate whether using the actual lowest error stump may improve the performance of boosting.

We ran 10 iterations of the Adaboost algorithm on several datasets, using both information gain (normal boosted stumps) and alternatively selecting the stump with the lowest error. We used 80% of the data from each dataset to train and the remaining 20% as a test. The results are summarized in Table 5, which display train and test set error rates of the ensemble over 10 iterations on five datasets.

We observe that of the 50 iterations, over 30 (60%) had a misordered feature selected by information gain. Further, in many cases, the ensemble built by choosing stumps with the lowest error also had better accuracy on these datasets during training and testing. While many factors affect performance in real datasets, this is an indication that reducing misordering may in some cases improve the performance of a downstream classifier or ensemble method.

### 4.4. Impact of Misordering on Feature Selection Precision and Recall

In this section, we evaluate the efficacy of various feature selection metrics, Information gain, Gini, Hellinger distance, and Bhattacharyya distance in identifying relevant features from a dataset.

We use two datasets, Madelon [44] and Gisette [45], from the NIPS 2003 feature selection challenge which contain a mixture of relevant and irrelevant features. Each dataset has a set of relevant features augmented with some irrelevant “probe” features. Relevant features for Madelon were extracted using the Boruta method [46], and their relevance was verified using correlation matrix analysis. For Gisette, we found a version of the dataset that identified the probe features explicitly. We then created a more challenging version we call Gisette-small by keeping all irrelevant features but only 500 of the original 2500 relevant features. We then run all feature selection methods on the two datasets. Since the relevant features are known, we plot the precision and recall of selected features using the known relevant features as the ground truth, at different thresholds of the number of selected features (“Top-n features”) on the *x*-axis. The results are shown in Figure 10 and Figure 11. In these results, the Gini index curve almost completely overlapped with the information gain curve (not surprising as their misordering rates are quite similar), so we did not plot the Gini result.

From the results for Madelon, Hellinger demonstrates a higher recall rate at a lower number of top-n features, followed by Bhattacharya distance and then information gain which shows higher recall at a larger value of top-n features. Bhattacharyya distance has the best overall recall. This trend correlates with the misordering rates (Table 6) which show higher misordering rates for Information gain followed by Bhattacharya distance and Hellinger distance. In Gisette-small, Bhattacharyya distance still has the best overall recall rate and matches the precision of information gain. This is consistent with its lower misordering rate than information gain. However, Hellinger distance does not perform well on this dataset though it has a low misordering rate on this dataset. The reason for this behavior remains to be understood. With this exception, however, the other results indicate that the misordering rate correlates well with true feature selection performance.

## 5. Limitations

In this study, we have made several assumptions to simplify our analysis. Our focus has been on binary class data with binary features for ease of analysis. Extension to multi-class problems remains for future work. While the approach should be extendable, the selection criteria we have explored may behave differently in that case. Further, our analysis has been primarily based on four feature selection metrics. There are many other feature selection metrics that remain to be analyzed using this approach. We also assumed that no data were missing. Missing feature values would not have allowed us to accurately calculate the variables *p* and *q* in our analysis. In the presence of missing values, we would suggest using techniques such as imputation before a feature selection criterion is applied. Finally, we have performed preliminary experiments to evaluate the effect of misordering on downstream classifiers. While we observed differences in performance, our experiments cannot conclusively suggest differences in misordering rates as the cause. Designing additional experiments to understand downstream effects remains a direction for future work.

## 6. Conclusions

In this paper, we have discussed a not-so-well-studied aspect of the relationship between feature selection metrics and accuracy, and misorderings. We have looked at some common feature selection methods including Information gain, Gini index, Hellinger distance, and Bhattacharya distance and their relationship to misordering. We have shown analytically as well as empirically that misordering is a phenomenon that affects several different feature selection metrics. It is not confined to a narrow range of accuracy values or feature selection metrics and it is not a function of numerical errors introduced during computation. Further, different metrics have different behavior with respect to the parameters that characterize partitions of the data. Empirically, we show that misordering also happens in real-world data. Beyond the feature selection metrics we have tested, performance correlations with other metrics suggest that many metrics will be prone to misordering as well.

## Figures and Tables

**Figure 1 entropy-25-01646-f001:**
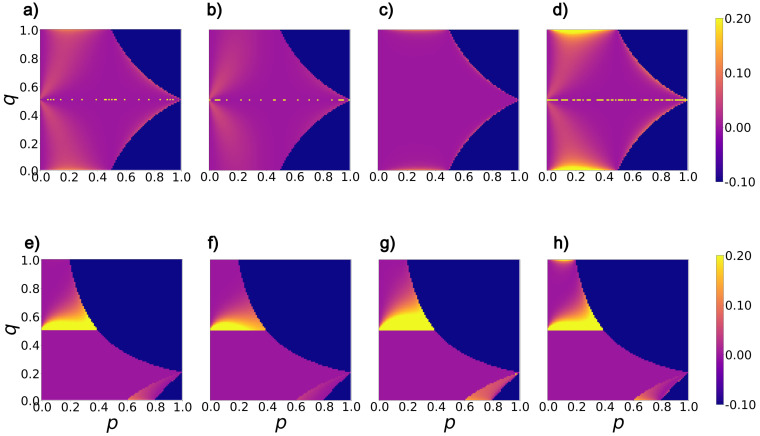
Heat maps illustrate the misalignment between the gradient vectors of feature selection metrics and accuracy. (**a**,**e**) depict this relationship for Information gain, (**b**,**f**) for the Gini index, (**c**,**g**) for the Hellinger distance, and (**d**,**h**) for the Bhattacharyya distance. (**a**–**d**) represent datasets with an equal distribution of negative and positive samples, while (**e**–**h**) show datasets with m=0.8. Here, *p* represents the fraction of examples in the first partition, and *q* represents the fraction of positive examples in the first partition.

**Figure 2 entropy-25-01646-f002:**
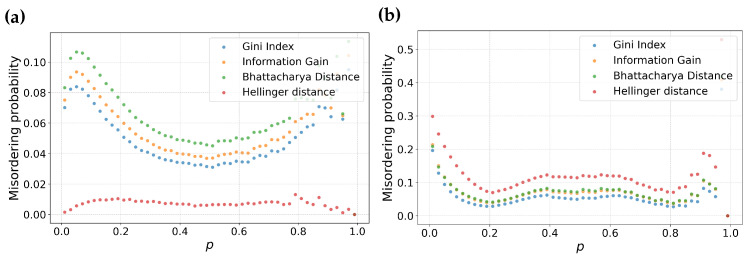
Probability of misordering for information gain, Gini index, Hellinger distance, and Bhattacharyya distance as functions of the parameters *p*. Sub-figure (**a**) represents these comparisons for balanced datasets, while sub-figure (**b**) shows a similar comparison for skewed datasets with m=0.8.

**Figure 3 entropy-25-01646-f003:**
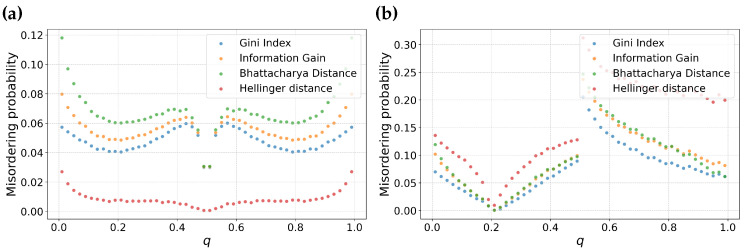
Probability of misordering for information gain, Gini index, Hellinger distance, and Bhattacharyya distance as functions of the parameters *q*. Sub-figure (**a**) represents the these comparisons for a balanced dataset, while sub-figure (**b**) shows similar comparison for a skewed dataset with m=0.8.

**Figure 4 entropy-25-01646-f004:**
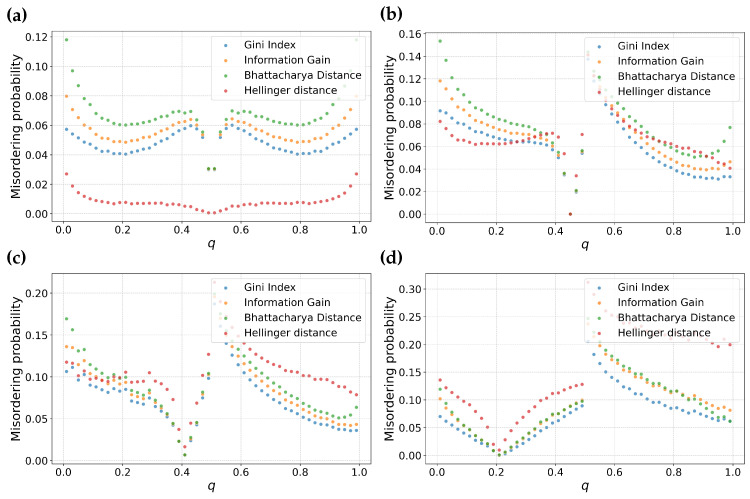
Probability of Misordering for Information Gain, Gini Index, Hellinger Distance, and Bhattacharyya Distance as a function of the Parameter *q*. This figure highlights the dependency of Hellinger Distance on data skew, with sub-figures (**a**–**d**) representing data distributions with varying fractions of negative samples (*m*): m=0.5 (**a**), m=0.55 (**b**), m=0.6 (**c**), and m=0.8 (**d**), respectively.

**Figure 5 entropy-25-01646-f005:**
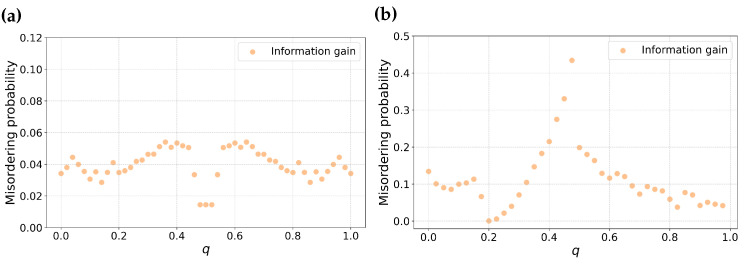
The empirical probability of misordering for Information gain as a function of the parameter *q*. Sub-figure (**a**) represents the empirical misordering probability for balanced datasets, while sub-figure (**b**) represents the same for skewed datasets with m=0.8.

**Figure 6 entropy-25-01646-f006:**
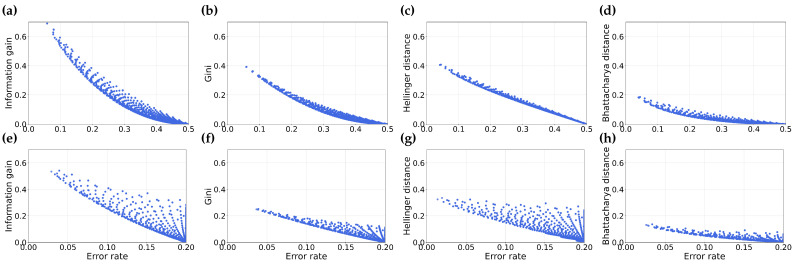
This figure illustrates the comparison between the error rate and various feature selection metrics—Information gain (**a**,**e**), Gini index (**b**,**f**), Hellinger distance (**c**,**g**) and Bhattacharyya distance (**d**,**h**). Sub-figures (**a**–**d**) represent datasets with equal distribution of negative and positive samples, while sub-figures (**e**–**h**) show the same comparisons for datasets with m=0.8.

**Figure 7 entropy-25-01646-f007:**
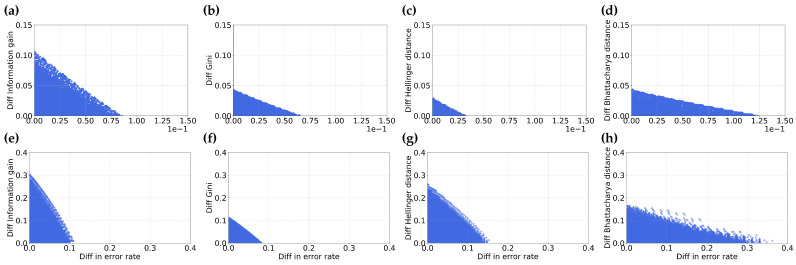
This figure illustrates the comparison between the difference in error rate and difference in various feature selection metrics–Information gain (**a**,**e**), Gini index (**b**,**f**), Hellinger distance (**c**,**g**) and Bhattacharyya distance (**d**,**h**). Sub-figures (**a**–**d**) represent datasets with equal distribution of negative and positive samples, while sub-figures (**e**–**h**) show the same comparisons for datasets with m=0.8.

**Figure 8 entropy-25-01646-f008:**
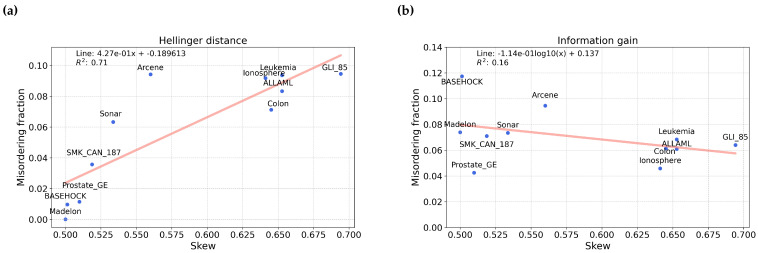
Relationship between the class skew in real datasets and the percent misordering for (**a**) Hellinger distance (**b**) Information gain. Each data point represents a unique dataset. A line is drawn to show goodness of fit with a linear model.

**Figure 9 entropy-25-01646-f009:**
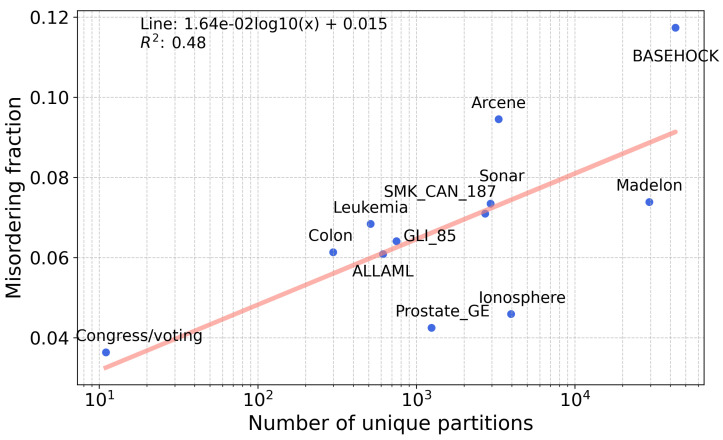
Relationship between the number of features in real datasets and the percent misordering. Each data point represents a unique dataset. A line is drawn to show goodness of fit with a linear model.

**Figure 10 entropy-25-01646-f010:**
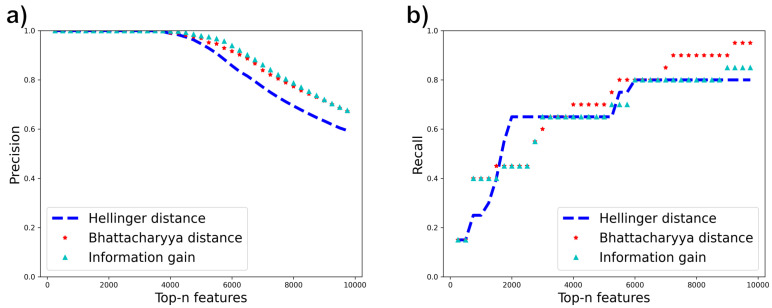
(**a**) Precision (**b**) Recall of relevant features using Information gain, Hellinger distance and Bhattacharyya distance for Madelon.

**Figure 11 entropy-25-01646-f011:**
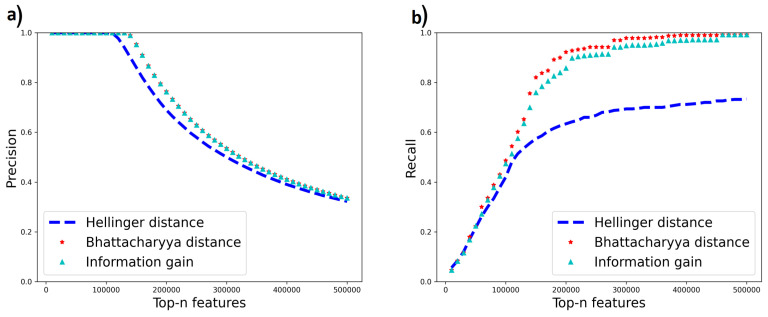
(**a**) Precision (**b**) Recall of relevant features using Information gain, Hellinger distance and Bhattacharyya distance for Gisette-small.

**Table 1 entropy-25-01646-t001:** Toy Classification Task to Predict Which Animals Are Lions.

Sharp Teeth?	Has Fur?	Lazy?	Loud?	Is Lion? (Label)
No	No	Yes	No	Yes
No	Yes	No	No	No
Yes	No	Yes	No	No
Yes	Yes	Yes	No	Yes
Yes	Yes	No	No	No
Yes	No	No	No	No
Yes	Yes	Yes	No	Yes
Yes	Yes	No	No	Yes
Yes	Yes	No	Yes	Yes

**Table 2 entropy-25-01646-t002:** Summary of Real Datasets.

Dataset	Percent Majority Class	Number of Examples	Number of Features	Number of Unique Feature Splits	Feature Type
Gisette	0.500	7000	5000	478,699	Continuous
Madelon	0.500	2600	500	29,490	Continuous
BASEHOCK	0.501	1993	4862	43,109	Discrete
Prostate_GE	0.510	102	5966	1245	Continuous
SMK_CAN_187	0.519	187	19,993	2715	Continuous
Sonar	0.534	208	60	2941	Continuous
Arcene	0.560	200	10,000	3308	Continuous
Congressional Voting	0.614	435	16	11	Binary
Ionosphere	0.641	351	34	3958	Continuous
Colon	0.645	62	2000	299	Discrete
ALLAML	0.653	72	7129	615	Continuous
Leukemia	0.653	72	7070	515	Discrete
GLI_85	0.694	85	22,283	749	Continuous

**Table 3 entropy-25-01646-t003:** Misordering Rates on real datasets for Information Gain, Gain Ratio, Gini Index, Hellinger distance, and Bhattacharyya distance.

Dataset	Percent Majority Class	Information Gain	Gain Ratio	Gini Index	Hellinger Distance	Bhattacharyya Distance
Gisette	0.500	0.0612	0.116	0.0572	0.0024	0.0562
Madelon	0.500	0.0739	0.143	0.0683	0.0001	0.0455
BASEHOCK	0.501	0.117	0.209	0.113	0.0097	0.1122
Prostate_GE	0.510	0.0425	0.0837	0.0325	0.0115	0.0593
SMK_CAN_187	0.519	0.0710	0.127	0.0629	0.0357	0.0805
Sonar	0.534	0.0734	0.113	0.0686	0.0633	0.0775
Arcene	0.560	0.0945	0.124	0.0851	0.0943	0.1037
Congressional Voting	0.614	0.0364	0.0182	0.0364	0.0364	0.0545
Ionosphere	0.641	0.0459	0.0263	0.0433	0.0918	0.0454
Colon	0.645	0.0614	0.0603	0.0490	0.0713	0.0716
ALLAML	0.653	0.0609	0.0564	0.0444	0.0833	0.0738
Leukemia	0.653	0.0684	0.0631	0.0551	0.0936	0.0793
GLI_85	0.694	0.0641	0.0480	0.0489	0.0946	0.0745

**Table 4 entropy-25-01646-t004:** Correlation of Various Feature Selection Metrics With Information Gain.

Feature Selection Method	Pearson’s Coef.	*p*-Value	# Data Points
Fisher score	0.961	1.22 × $10^{-12}$	22
CFS (Correlation Based Feature Selection)	0.929	6.70 × $10^{-93}$	213
INTERACT	0.913	9.98 × $10^{-37}$	92
Cons (Consistency Based Filter)	0.912	1.11 × $10^{-73}$	187
Chi-squared statistic	0.899	1.81 × $10^{-35}$	96
Relief-F	0.861	8.27 × $10^{-72}$	240
mRMR	0.851	5.30 × $10^{-21}$	71
FCBF (fast correlation-based filter)	0.836	3.68 × $10^{-36}$	134
LASSO	0.726	5.89 × $10^{-16}$	90
SVM-RFE	0.660	3.98 × $10^{-19}$	142

**Table 5 entropy-25-01646-t005:** Train and test set accuracy of Adaboost with decision stumps chosen by information gain (IG/Train and IG/Test) compared with stumps chosen by minimum error (Err/Train and Err/Test). Bold iterations represent where misordering occurs when using IG.

Dataset	Iteration	IG/Train	Err/Train	IG/Test	Err/Test
Volcanoes [42]	1	0.841	0.841	0.029	0.029
	2	0.841	0.841	0.029	0.029
	**3**	0.841	0.841	0.029	0.029
	4	0.845	0.855	0.166	0.126
	**5**	0.845	0.855	0.166	0.126
	6	0.845	0.842	0.164	0.235
	**7**	0.839	0.866	0.211	0.161
	**8**	0.846	0.867	0.155	0.191
	**9**	0.840	0.864	0.211	0.112
	**10**	0.846	0.871	0.155	0.197
Spam [43]	**1**	0.663	0.712	0.668	0.707
	**2**	0.663	0.712	0.668	0.707
	**3**	0.663	0.712	0.668	0.707
	4	0.675	0.706	0.677	0.714
	5	0.721	0.715	0.715	0.716
	6	0.721	0.717	0.715	0.718
	**7**	0.721	0.719	0.715	0.713
	**8**	0.711	0.722	0.706	0.720
	9	0.719	0.720	0.714	0.713
	**10**	0.719	0.723	0.714	0.721
Ionosphere [22]	1	0.864	0.864	0.700	0.700
	2	0.864	0.864	0.700	0.700
	**3**	0.914	0.907	0.886	0.771
	4	0.893	0.907	0.771	0.714
	**5**	0.893	0.904	0.771	0.743
	**6**	0.896	0.929	0.771	0.786
	**7**	0.925	0.907	0.800	0.786
	**8**	0.896	0.932	0.771	0.800
	**9**	0.943	0.932	0.800	0.786
	**10**	0.911	0.946	0.771	0.857
Hepatitis [22]	1	0.889	0.889	0.875	0.875
	**2**	0.889	0.889	0.875	0.875
	3	0.889	0.905	0.875	1.000
	4	0.905	0.921	1.000	1.000
	5	0.857	0.952	1.000	1.000
	6	0.952	0.968	1.000	1.000
	7	0.984	0.968	1.000	1.000
	**8**	0.968	0.968	1.000	1.000
	9	0.984	0.984	1.000	1.000
	10	0.952	0.984	1.000	1.000
German [22]	**1**	0.700	0.715	0.680	0.725
	2	0.700	0.715	0.680	0.725
	**3**	0.723	0.730	0.735	0.710
	4	0.717	0.741	0.715	0.740
	**5**	0.715	0.742	0.715	0.745
	**6**	0.715	0.750	0.715	0.740
	**7**	0.725	0.751	0.735	0.725
	**8**	0.736	0.761	0.715	0.730
	**9**	0.757	0.770	0.715	0.740
	**10**	0.762	0.760	0.715	0.730

**Table 6 entropy-25-01646-t006:** Misordering rates for Madelon and Gisette-small.

Dataset	Information Gain	Hellinger Distance	Bhattacharyya Distance
Madelon	0.0739	0.0001	0.045
Gisette-small	0.0534	0.0007	0.034

## Data Availability

Code and data will be made available on github upon publication.

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
