# Peer review of "On the Relationship between Feature Selection Metrics and Accuracy"

_entropy, 2023, doi:10.3390/e25121646_

Round 1

Reviewer 1 Report

Comments and Suggestions for Authors

The article addresses the link between classification quality and feature selection measures and differences in feature ranking order. I think the topic may be interesting, but there is no clear justification of the subject matter. In fact, why do it? Is it really such a problem, or does it have such a big impact on the quality of the resulting models? Please justify, maybe some concrete example, such as Madelon base, show the ranking and position differences and their interpretation. Maybe show a specific action on this base.

Some comments and additional questions:
- why were these four measures of feature selection chosen and what about the others?
- how were the data sets divided into training and test sets?
- how was accuracy calculated?
- I suggest making a graphical diagram of the proposed approach.
- please add clear examples of missorderings, and how is accuracy calculated in assessing missordering?
- what tools and algorithm implementations were used in the experiments?
- line 14 numbered from 0?
- figure 1 two times marks a,b,c,d.

Comments on the Quality of English Language

Please check for minor spelling errors.

Reviewer 2 Report

Comments and Suggestions for Authors

Reviewer 3 Report

Comments and Suggestions for Authors

This article offers a comprehensive analysis of the discrepancies between common feature selection metrics and model accuracy in the realm of machine learning. The focus on 'misorderings'—situations where the rankings provided by feature selection metrics diverge from those suggested by accuracy—is particularly insightful. The authors' methodical approach, blending analytical investigation with empirical evaluation, lends credibility to their findings. They explore the frequency of misorderings across various metrics as influenced by parameters related to data partitioning. This analytical part is robust, providing a theoretical foundation for understanding systematic differences among metrics. Furthermore, the empirical evaluation using real-world datasets not only corroborates the analytical findings but also grounds the study in practical relevance. Overall, the research significantly contributes to our understanding of feature selection metrics, although it leaves some questions about practical applications and broader impacts unanswered.

Major comments;

- However, the paper could benefit from a deeper exploration of the implications of these misorderings on final model performance and potential strategies to mitigate their impact. 

- Likewise, a discussion on the applicability of these findings across different types of machine-learning problems would enhance the paper's utility for a broader audience. Specifically, if you could summarize the most common classification algorithms in sci-kit learn, their default metric for evaluating feature importance, and a suggestion whether for that algorithm it is recommended to "tune" this metric for this classifier as a hyperparameter could be valuable.

- Another practical comment that you might consider when reporting the misorderings. If features A and B have really similar importance per some metric, it maybe is irrelevant if A is more important than B, or the other way around. This difference could stem from a particular train/test split, and the order could flip if the experiment was repeated. So, having this in mind, maybe it's better to evaluate whether the features (or, in fact, the unique splits as you aptly describe) are ranked in the same feature importance percentile or not and not comparing the feature importances directly. Maybe this would mitigate these irrelevant "misorderings" and provide more robust insights.

Minor remarks:

The introduction has a 0 numbering before it; please fix the typo.

- How would you address or what is the impact if the assumption in line 70 does not hold ("no feature values are missing for analytical comparisons")?

- The description you provide in lines 92-96 about p and q could also be briefly repeated somewhere closer to Figure 1 (maybe even in the caption) for easier interpretation of the heatmaps.

- In section 1.1, clarify which data set the empirical evaluation is about.

Comments on the Quality of English Language

n/a

Round 2

Reviewer 1 Report

Comments and Suggestions for Authors

we can accept this version of paper

Comments on the Quality of English Language

we can accept this version of paper

Reviewer 2 Report

Comments and Suggestions for Authors

The authors have attended all comments of this reviewer.

Reviewer 3 Report

Comments and Suggestions for Authors

The authors have addressed some of my concerns. The others are postponed to a later work.